# Influence of Environmental Exposure to Steel Waste on Endocrine Dysregulation and *PER3* Gene Polymorphisms

**DOI:** 10.3390/ijerph20064760

**Published:** 2023-03-08

**Authors:** Gilvania Barreto Feitosa Coutinho, Maria de Fátima Ramos Moreira, Frida Marina Fischer, Maria Carolina Reis dos Santos, Lucas Ferreira Feitosa, Sayonara Vieira de Azevedo, Renato Marçullo Borges, Michelle Nascimento-Sales, Marcelo Augusto Christoffolete, Marden Samir Santa-Marinha, Daniel Valente, Liliane Reis Teixeira

**Affiliations:** 1Center for Studies on the Worker’s Health and Human Ecology, Sergio Arouca National School of Public Health, Oswaldo Cruz Foundation, 1480 Leopoldo Bulhões St., Rio de Janeiro 21041-210, RJ, Brazil; 2Department of Environmental Health, School of Public Health, University of São Paulo, São Paulo 01246-904, SP, Brazil; 3Center for Natural and Human Sciences (CCNH), ABC Federal University (UFABC), Santo André 09210-580, SP, Brazil; 4Centro de Ciências Biológicas e de Saúde (CBS), Universidade Cruzeiro do Sul (Unicsul), São Paulo 01506-000, SP, Brazil

**Keywords:** steel industry, environmental exposure, endocrine disruptors, *PER3* gene polymorphism, chronotypes

## Abstract

Objective: To evaluate the association between environmental exposure to the following chemical substances: cadmium (Cd), lead (Pb), nickel (Ni), manganese (Mn), benzene (BZN), and toluene (TLN), and Period Circadian Regulator 3 (*PER3*) gene variable number of tandem repeats (VNTR) polymorphisms, according to chronotype in a population living in a steel residue-contaminated area. Methods: This assessment comprises a study conducted from 2017 to 2019 with 159 participants who completed health, work, and Pittsburgh sleep scale questionnaires. Cd, Pb, Ni, Mn, BZN, and TLN concentrations in blood and urine were determined by Graphite Furnace Atomic Absorption Spectrometry (GFAAS) and Headspace Gas Chromatography (GC), and genotyping was carried out using Polymerase Chain Reaction (PCR). Results: A total of 47% of the participants were afternoon chronotype, 42% were indifferent, and 11% were morning chronotype. Insomnia and excessive sleepiness were associated with the indifferent chronotype, while higher urinary manganese levels were associated with the morning chronotype (Kruskal–Wallis chi-square = 9.16; *p* < 0.01). In turn, the evening chronotype was associated with poorer sleep quality, higher lead levels in blood, and BZN and TLN levels in urine (χ^2^ = 11.20; *p* < 0.01) in non-occupationally exposed individuals (χ^2^ = 6.98; *p* < 0.01) as well as the highest BZN (χ^2^ = 9.66; *p* < 0.01) and TLN (χ^2^ = 5.71; *p* < 0.01) levels detected in residents from the influence zone 2 (far from the slag). Conclusion: Mn, Pb, benzene, and toluene contaminants may have influenced the different chronotypes found in the steel residue-exposed population.

## 1. Introduction

Human populations living near steel plants are environmentally exposed to residues [1,2,3,4,5,6]. The steel industry is characterized as a transformation industry for steel production because it involves the modification of raw materials such as charcoal or coke, sinter, and pellet iron ores. Other materials include quartz and limestone for the production of blast furnace gas, pig iron, steel slag, dust, and mud. This process generates steel slag as waste, containing calcium and metals such as silicon (Si), aluminum (Al), iron (Fe), manganese (Mn), magnesium (Mg), cadmium (Cd), chromium (Cr), lead (Pb), and nickel (Ni) [7,8].

In addition, benzene and toluene are two of the twenty most commonly used volatile organic compounds in industrial production [9,10], and they are utilized by the petrochemical and steel industries [8,11]. Exposure to these solvents can occur in two ways: when burning petroleum products or when industrial waste accumulates in deposits [12].

Such industrial waste pollutes the soil, air, surface and groundwater, and food, posing a health risk to those who are exposed. This is because these contaminants can harm many organs and systems, including the endocrine system, and thus affect their biological functions [4,13,14,15,16].

Long-term environmental exposure to low metal concentrations is the most common, and it is difficult to establish a cause–effect relationship because the effects of contamination take years to manifest and are generally nonspecific. Furthermore, each metal has its own toxicodynamic and toxicokinetic properties, such as molecular mimetism, oxidative damage, and bonds with DNA and proteins, which are also common toxicity mechanisms [17]. Lead toxicity primarily affects the nervous and hematopoietic systems, kidneys, and gastrointestinal tract in humans. It also has an impact on reproduction and development, as well as having a negative impact on the cardiovascular system. Ni inhalation induces cancer by affecting the respiratory system, nasal and mucosa cavity, kidneys, liver, and brain. Furthermore, Ni ingestion has been linked to gastrointestinal, hematological, hepatic, renal, and neurological problems, and dermal contact with Ni can cause dermatitis. In addition to being carcinogenic, hepatotoxic, and genotoxic, Cd causes hormonal dysfunctions and damages the kidneys, cardiovascular, respiratory, hematological, and nervous systems. Manganese is an essential element for humans, as it is involved in bone and tissue formation, reproductive functions, and carbohydrate and lipid metabolism. However, at high concentrations, such as those found in workplaces, it is neurotoxic [17].

Given the volatility and widespread use of benzene and toluene, inhalation is the most common route of exposure for these contaminants [18,19]. Human health consequences include central nervous system and auditory effects, changes in renal and dermatological functions, and, most importantly, changes in hormone levels [10,20,21,22,23,24]. Benzene primarily damages the bone marrow, which can result in a variety of hematological changes, including hypoplasia, dysplasia, and aplasia, as well as cancer [25]. Furthermore, rat studies have identified sleep disorders as one of the toluene and benzene exposure symptoms [23,26,27,28,29].

Aside from contaminating the environment, such toxics also act as Endocrine Disrupting Compounds (EDCs), as they can alter the endocrine system functions [15,17,30], which are considered the main organism interface with the environment [13,31]. As a result, hormonal disorders caused by endocrine gland dysfunction can result in a wide range of diseases caused by both acute and chronic exposure, including sleep disorders caused by the pineal gland’s inappropriate release of melatonin. This one compound is the primary endocrine system hormone, which regulates circadian rhythms, including the rhythmicity of the sleep–wake cycle, among other functions [13].

The principal clock genes (*CLOCK*, *BMAL1*, *CRY1*, *CRY2*, *CKΕ*, *CKΣ*, *PER1*, *PER2*, and *PER3*) control the circadian rhythms in mammals. Changes in those genes can result in a variety of circadian genotypes, such as the morning/evening character, due mainly to interferences in melatonin expression and synthesis control [32,33,34,35], which cause rhythmicity disorders in the sleep–wake cycle [13]. Because the *PER3* gene VNTR polymorphism affects sleep–wake cycle synchronism [36], endocrine-disrupting substances (manganese, lead, benzene, and toluene) can modify the active sites in the central nervous system, where the pineal gland is located [37]. In addition, they may be neurotoxic through the hypothalamic–pituitary–adrenal axis, which stimulates the secretion of adrenocorticotropic hormone (ACTH), which is responsible for cortisol synthesis and secretion, besides initiating a melatonin inhibition or release response [38,39].

The association between environmental exposure to such substances and the sleep–wake cycle requires investigation. There are few studies on the subject, especially relating metals with *PER3* gene VNTR polymorphism. This study aimed to investigate the association between environmental exposure to chemical substances (Cd, Pb, Ni, Mn, benzene, and toluene) and *PER3* gene VNTR polymorphisms according to chronotype in a population residing in a steel residue-contaminated area.

## 2. Materials and Methods

A cross-sectional study was conducted in an adult population in Volta Redonda, Rio de Janeiro, Brazil. The study’s subjects live near a steel waste dump. Furthermore, the condominium was built in an area that was previously used as a steel waste disposal site. The data were collected between July 2017 and January 2019. A convenience sample of the population was taken. Participants included 203 residents who had lived at the site for more than six months. Fourteen people who used medication for sleep disorders and thirty people who had some sleep disorders were excluded, leaving one hundred and fifty-nine people.

Study area

The condominium was divided into two zones of influence (Figure 1), with each street’s roundabout serving as the boundary between the two zones. Zone of influence 1 (near the slag) is near the steel slag wall, whereas Zone of Influence 2 (away from the slag) is near the main street (1043 St.). The distance between the wall and the roundabout is about 140 m, and the complementary space to the main street is about 252 m. The chosen location connects the condominium’s internal regions and allows for population movement.

Data collection

During the first stage, the study participants answered two questionnaires: (1) a comprehensive questionnaire including information on working conditions and sociodemographic characteristics (age, marital status, education, and employment information, i.e., working time and chemical exposure to metals at work), lifestyles (use of tobacco, alcohol, and the consumption of other psychoactive substances), physical activity, familiar disease history, past and current health conditions. (2) The Pittsburgh Sleep Quality Index (SQI) scale developed by Buysse and collaborators [40] was also answered, comprising scores on quality and patterns of sleep.

Biological samples

The second stage of the study comprised biological sample collection (urine and blood) to investigate Cr, Cd, Ni, Pb, Mn, benzene, and toluene exposures.

Urine samples were used for Cr, Ni, Cd, and Mn analysis. Whereas the collection process is non-invasive and thus more acceptable for obtaining samples, urine is one of the most-used biomarkers in the case of chronic exposure [41]. Cd accumulates in the kidneys and other tissues, and urinary Cd represents this total body content, reflecting a critical exposure [42]. Ni is an essential metal for human body physiology. However, there are currently no adequate exposure biomarkers. Because there is little correlation between environmental exposure and individual urine levels, group assessment is the most weighted way to estimate population risk [43]. Mn is also an essential element. Although it can be measured with high sensitivity in several biological compartments, including blood and urine, no adequate exposure biomarker exists [44]. Finally, Pb is stored in the bones, with less than 2% found in the blood. Although bone Pb is regarded as a chronic exposure biomarker, it is not uniformly distributed, accumulating in bone regions where calcification is most active during the exposure period. As a result, the most appropriate biomarker would be plasma Pb, which is directly related to lead in bone but is very difficult to measure due to hemolysis and its levels being close to the quantification limits of most analytical techniques. Given all these reasons, BPb remains the biomarker of choice for lead exposure around the world [45].

Metal determinations were carried out in two atomic absorption spectrometers, A Analyst 800 and 900 (Perkin Elmer, Norwalk, CT, USA), equipped with transverse electrothermal atomizers, longitudinal Zeeman background correctors, and AS-800 automatic samplers (Perkin Elmer, Norwalk, CT, USA). The methodologies used followed the previously established protocols by the Metals Sector in the Toxicology Laboratory (Cesteh/Ensp/Fiocruz). Whole blood was collected in heparinized vacutainer tubes for trace metal analyses, while urine was collected in previously decontaminated 50 mL containers. Cr, Ni, and Cd were determined in urine, Mn in urine and blood, and Pb in blood.

The determination of unmetabolized urinary benzene and toluene employed the solid phase microextraction technique (SPMT), an Agilent headspace gas chromatograph, and an autosampler. Ions 78 and 91 *m/z* were used for benzene and toluene quantification, respectively.

In the third stage, DNA was extracted from blood samples using the salting out method, according to Miller et al. [46]. After isolation, the genotyping of the number from the *PER3* gene VNTR polymorphism associated with different chronotypes [33,47] was performed by conventional polymerase chain reaction (PCR) technique, according to Pereira et al. [34,47]. Reaction was carried out in MasterCycler thermocycler (Eppendorf AG, Hamburg, Germany) for 95 °C–2 min (HotStart), followed by 95 °C–15 s, 60 °C–25 s, and 72 °C–30 s for 35 cycles. We used 10 ng of DNA, 1× PCR Buffer, 0.2 mM dNTP mix, 0.33 μM primer mix and 5U *Taq* DNA Polymerase accordingly to manufacture (Sigma-Aldrich Inc., St Louis, MO, USA) standard protocol. Sense primer 5′-GAGCAGTCCTGCTACTACCG-3′ and antisense 5′-CTTGTACTTCCACATCAGTGCC-3′ were used. PCR products were submitted then to electrophoresis in Ethidium Bromide stained 1% agarose gel at 90 V for 1 h and image from the gel acquired with SmartView Pro Imager System (Major Science Co., Ltd., Taoyuan City, Taiwan). PCR products were 324 bp, 378 bp or both, corresponding to *PER3*^4/4^ (evening), *PER3*^5/5^ (morning) and *PER3*^4/5^ (intermediate) chronotypes, respectively (Figure 2). For allelic frequency and Hardy–Weinberg equilibrium (HWE), all 4 repeats were considered as common allele (p) and 5 repeats were considered as rare allele (q).

Statistical analyses

The descriptive data analysis first computed the means, standard deviations, and medians for all continuous variables, as well as the absolute and relative frequencies for categorical data. The data were then checked for normality (Shapiro–Wilk test). Because the data were not normally distributed, the Kruskal–Wallis chi-square test and the Wil-coxon test were used. A linear multivariate analysis was carried out controlled for potential confounders such as gender, age group, work exposure time, and/or residence time in a steel-residues contaminated area. It was not possible to control smoking and drinking habits, due to the small number of subjects. For Deviations from HWE in *PER3* VNTR polymorphism frequencies were assessed by goodness-of-fit χ^2^ test, considering 3.841 as critical value, degree of freedom 1 and α = 0.05. All analyses were set at a significance level of *p* = 0.05 and a confidence interval of 95% (CI = 95%). The Bar Charts with the median and the 95% confidence interval around this median were made. The Statistical Package for the Social Sciences 17.0 for Windows^®^ software (SPSS Inc., Chicago, IL, USA) was used for all statistical assessments.

## 3. Results

### 3.1. Sociodemographic Aspects and PER3 Gene VNTR Polymorphism Frequencies

Of the 159 subjects, 69 (44%) were men and 90 (56%) were women, aged between 18 and 86 (median = 51 years old) and living in the condominium between 1 and 30 years (median = 14 years). The participants were mostly ex-smokers (18.5%), with only a small percentage (7.8%) claiming to be smokers. Regarding drinking habits, 37.6% drank only at social gatherings, and 4.8% had previously consumed alcohol. The study population included 43 people who reported doing domestic activities (23%), nine students (5%), and 19 retirees (10%). Of the 42 workers, all reported chemical or physical exposure at work (22.2%), with exposure periods ranging from 1 to 48 years (median = 10 years). Of 159 residents genotyped, 75 (46.9%) were *PER*^4/4^, 67 (41.8%) were *PER*^4/5^ and 17 (11.3%) *PER*^5/5^. Allelic frequency for 4 repeats was 66.5% and for 5 repeats 33.5%. For HWE, p^2^ = 0.442 (4 repeats, expected 70), 2pq = 0.446 (expected 71) and q^2^ = 0.112 (5 repeats, expected 18) Goodness-of-fit χ^2^ test is 0.638, below 3.841 critical value, showing no deviation from HWE. The highest prevalence of the evening chronotype was observed for both men and women (women 48.4%, men 51.5%). The highest participation was of individuals over 50 (50.8%), 42.9% were between 22 and 50, and only 4.8% were under 21. The indifferent chronotype comprised 57.1% of ages below 21, while the evening chronotype included 54.4% of those over 50. According to habits, 41.7% of the afternoon chronotypes were smokers, and 57% did not drink alcohol.

### 3.2. Sleep–Wake Cycle According to PER3 Gene Polymorphism in the Population

The sleep patterns according to *PER3* gene VNTR polymorphisms are in Table 1. Sleep latency in the morning chronotype group was lower (median = 10 min) when compared to the indifferent and evening (median = 15 min) chronotypes. When it came to bedtime, there were similarities between the morning, indifferent, and evening groups (median = 11 PM). The wake-up time was earlier for the morning group (median = 6 AM) than the other two groups (median = 6:30 AM). The sleep duration was slightly greater among the indifferent and evening groups (median = 8 h) compared to the morning group (median = 7 h). Excluding bedtime, morning chronotypes differed from the evening and indifferent groups for the other analyzed variables, albeit with no statistical significance.

### 3.3. Sleep Complaints According to PER3 Gene VNTR Polymorphism in the Population

Table 2 shows sleep complaints according to *PER3* gene polymorphisms. The indifferent group had a higher number of complaints about insomnia (58%) than the evening (32%) and morning (10%) groups (Friedman’s chi-square = 154; *p* < 0.01). The same pattern was observed for excessive sleepiness (60%) in the indifferent group compared to the evening (36%) and morning (4%) groups (Friedman’s chi-square = 158; *p* < 0.01). The evening group (55%) had more complaints about poor sleep quality than the indifferent (38%) and morning (7%) groups (Friedman’s chi-square = 161; *p* < 0.01).

### 3.4. Lead, Manganese, Cadmium, and Nickel Levels in Blood and Urine According to PER 3 Gene VNTR Polymorphisms

Table 3 presents the concentrations of the metals investigated in blood and urine according to chronotypes. Table 4 and Table 5 show lead and manganese levels in blood according to chronotype, stratified by age group, drinking habits, and chemical exposure at work. Figure 3, Figure 4, Figure 5, Figure 6 and Figure 7 show median values for those metals according to subjects’ chronotype.

#### 3.4.1. Lead in Blood (BPb)

Although BPb has borderline significance, concentrations in the evening group (2.05 µg dL^−1^) were higher than in the indifferent and morning groups (Table 3). Concerning the age group and *PER3* gene polymorphisms (Table 4), BPb levels were higher in the evening group in individuals above 50 years. As for drinking habits, the highest BPb levels were detected in the evening group (2.26 µg dL^−1^), especially among those who reported current drinking (2.60 µg dL^−1^). For chemical exposure at work, BPb was higher in the evening group (1.91 µg dL^−1^) and lower in the morning group (1.26 µg dL^−1^).

#### 3.4.2. Manganese in Blood (BMn)

Table 5 presents BMn according to chronotype for sex, age group, smoking and drinking habits, and chemical exposure at work.

Female subjects showed higher levels when compared to men (W = −2.39; *p* = 0.02), having the highest observed median level among indifferent group (8.90 µg L^−1^). In terms of the subjects’ ages, BMn levels decrease as they get older (Kruskal–Wallis χ^2^ = 6.81; *p* = 0.03). Individuals under the age of 21 had the highest concentrations in the evening groups (9.61 µg L^−1^), while those over the age of 50 had the lowest concentrations (6.64 µg L^−1^). Smoking and drinking habits had increased BMn levels among morning subjects, yet with no statistical significance. Similarly, chemical exposure at work had no effect on the analyzed substance.

#### 3.4.3. Cadmium, Nickel, and Manganese in Urine (UCd, UNi, UMn)

Cd levels in urine were comparable across *PER3* gene VNTR polymorphisms, with no statistical differences.

Despite having the highest Ni levels in the morning group, no statistical significance was found between groups.

Higher Mn concentrations in urine were observed in the morning group (0.64 µg g^−1^ creat) compared with the evening and indifferent groups (0.44 µg g^−1^ creat and 0.35 µg g^−1^ creat, respectively) (Kruskal–Wallis chi-square = 9.16; *p* < 0.01) (Figure 3).

#### 3.4.4. Linear Regression Analysis According to PER 3 Gene VNTR Polymorphisms

The linear regression analysis indicated that Mn in urine was the only metal associated with the *PER3* gene VNTR polymorphism (β = 0.39 µg g^−1^ creat; CI > 0 and *p* < 0.01). According to the multivariate linear regression (Table 6), the highest Mn levels in urine were associated with the morning group. The linear regression analysis for the chronotypes control variables were sex, age, time of residence, and time of exposure at work. It was not possible controlling for smoking and drinking habits.

### 3.5. Urinary Benzene e Toluene (µg L^−1^) According to PER3 Gene Polymorphisms

Table 7 presents the concentrations of unmetabolized benzene and toluene in urine according to chronotype. When compared to the indifferent and morning groups, the evening group had the highest levels of unmetabolized benzene (179.04 µg L^−1^) and toluene (148.58 µg L^−1^) with statistical significance (*p* < 0.01). Figure 8 and Figure 9 show the compounds median values according to subjects’ chronotype.

#### 3.5.1. According to *PER3* Gene VNTR Polymorphisms and Influence Zone

Table 8 displays the concentrations of unmetabolized benzene and toluene in the urine according to the *PER3* gene and the zone of influence. Benzene concentrations in individuals presenting the afternoon chronotype residing in zone 2 were higher (387.28 µg L^−1^; χ^2^ = 9.66; *p* < 0.01) compared to those in zone 1. However, there was no association between toluene concentrations in resident individuals presenting the afternoon chronotype in the two zones, despite the higher levels detected in individuals presenting the afternoon chronotype in Zone 2 (164.00 µg L^−1^; χ^2^ = 5.71; *p* < 0.01).

#### 3.5.2. According to *PER3* Gene VNTR Polymorphisms and Occupational Exposure to Chemicals

The concentrations of unmetabolized benzene and toluene in urine (µg L^−1^) are presented in Table 9 according to chronotype and occupational exposure. The benzene concentration (219.91 µg L^−1^) was significantly higher in unexposed individuals with the afternoon chronotype than in those with exposure at work (115.51 µg L^−1^) (Kruskal–Wallis χ^2^ = 11.20; *p* < 0.01), while there was no statistical significance for toluene.

## 4. Discussion

Associations between the *PER3*^4/4^ with higher BPb, and *PER*^5/5^ with higher UMn were observed in the study population. The indifferent chronotype was associated with sleep complaints (insomnia and excessive sleepiness), while higher levels UMn were observed in the morning chronotypes. The evening chronotype was associated with poorer sleep quality and higher BPb, including in exposed workers over 50 years of age. We also observed an association of *PER3* gene VNTR polymorphisms with higher levels of non-metabolized benzene and toluene in the urine of evening chronotype residents in zone 2 (distant from slag) and those unexposed at work.

### 4.1. Sociodemographic Aspects

In the present study, the *PER3* polymorphism associated with the evening chronotype (*PER3*^4/4^) was detected in approximately half of the study population, followed by the indifferent (*PER3*^4/5^), and morning (*PER3*^5/5^) chronotypes. The same association (*PER3* ^4/4^) was observed in previous research [48,49,50,51] using a validated questionnaire. On the other hand, two studies assessing *PER3* VNTR polymorphism [27] and a validated questionnaire [52], both concerning the Brazilian population, did not have results similar to a South African population [53,54]. Such discrepancies may be due to different latitudes, as photoperiod and ambient temperature variations in tropical countries throughout the year are low [55].

Due to the low participation rate of the resident population in the study, which made determining a significant association difficult, it is important to note that most participants have a social drinking habit, similar to a study of workers exposed to lead (Pb) in Taiwan, Asia [56], which reported a 38% alcohol consumption rate. However, the highest Pb concentrations were found among evening group drinkers, which was most likely due to a confounding variable. A similar finding was reported in a study [50], which also found associations between *PER3* gene VNTR polymorphisms afternoon chronotypes and alcohol abuse.

### 4.2. Sleep–Wake Cycle According to PER3 Gene VNTR Polymorphisms in the Population

Although there was no statistical significance, the results found were similar to those in the literature for both sleep latency and wake-up time, while sleep duration was shorter in the morning chronotype group than the others. Some studies in different populations around the world found that clock gene polymorphisms are independently associated with circadian phenotypes. Those authors found *PER3* gene VNTR polymorphism was associated with morning–evening tendencies [33,34]. Like our research, Carrier and collaborators found individuals identified as morningness prefer to wake up early in the morning compared to eveningness individuals [57]. Moreover, Taillard and colleagues found out late sleeping and waking especially on the weekends characterized eveningness individuals [58].

### 4.3. Sleep Complaints According to PER3 Gene VNTR Polymorphisms

Our study found statistical significance for insomnia complaints in people with an indifferent chronotype. In another study, conducted in Finland [59], insomnia complaints were observed for the afternoon chronotype. Because light sensitivity differs according to the *PER3* gene VNTR polymorphism, with *PER3*^5/5^ being more sensitive to light in the blue wavelength range than *PER3*^4/4^ [60], those differences may be associated with different latitudes.

Several studies [15,56,59,60] investigated the effects of metal exposure on human health and found that environmental exposure to these contaminants can cause potentially toxic effects and insomnia-related disorders, among other findings. Some research has concluded that toluene exposure causes insomnia [27,29,61], as well as in chemical dependents exposed to both toluene and benzene [23]. Even at low levels of exposure, toxic effects can cause oxidative stress and subsequent DNA damage, according to studies [62,63,64].

In addition, an association with excessive sleepiness (60%) in indifferent chronotypes was also observed in our study. Researchers [52] identified higher rates of excessive sleepiness in individuals presenting the afternoon chronotype when applying a validated questionnaire, while another study [59] observed the same association when employing a non-validated questionnaire. However, excessive sleepiness is associated with other factors, such as sleep duration and working hours, making further assessments difficult. The worse reported sleep quality (55%) observed in this study was for the afternoon chronotype. Exposure to metals, including Cd, Pb, Ni, and Mn, has been reported as able to impair sleep quality [52,56]. Mohammadyan et al. [65], when adjusting the analysis for age, work experience, body mass index, and exposure to Pb as revealed by blood concentrations, detected a significant relationship with poorer sleep quality. Some studies have identified that benzene and toluene exposures can lead to sleep disorders with intense motor activity and, consequently, worse sleep quality [23,28,29]. The central nervous system is one of the first areas to account for the toxicity of those solvents [10,20,21,22,24].

### 4.4. Assessment of Manganese in Urine and Lead in Blood Levels According to PER3 Gene VNTR Polymorphism

Metals are well known endocrine-disrupting compounds [13,15,30,59]. Even with environmental and occupational regulations concerning exposure to those elements, low-levels exposure can also interfere with the endocrine system and circadian rhythm control and maintenance at a molecular level [55]. In this regard, melatonin and its metabolites act as metal-chelating agents and play an important role in inhibiting oxidative stress [37].

Although borderline significance between chronotypes, individuals presenting higher BPb belonged to the evening chronotype, according to our findings. However, when the subjects’ ages were considered, Pb concentrations were higher in those over 50, since this toxic metal accumulates in blood [66].

Bone loss accelerates after menopause and bone demineralization may release bone lead into circulation. Osteoporosis and atherosclerosis may result from elevated homocysteine concentrations. In individuals over 50 years of age, BPb and homocysteine concentrations correlate as an increase in BPb leads to an increase in homocysteine levels [67]. In this way, some diseases, and physiological events, such as osteoporosis and menopause, may cause lead to be released from the bone into the blood [67,68].

Another result with borderline significance was that BPb decreased according to chronotype among workers who reported chemical exposure to metals, with higher levels found in subjects within the afternoon chronotype and lower levels found in subjects within the morning chronotype. In addition to communities living near industrial activities, occupational groups may face serious risks from exposure to that metal [13], as reported in another study that found an association between insomnia and Pb concentration in the urine of exposed workers [56]. The highest Pb levels are probably associated with the afternoon chronotype as the metal exhibits affinity to melatonin and can occupy its active site since a binding affinity exists between Pb and melatonin [38].

Elevated levels of manganese in urine were observed in the morning group in our study. Another research reported the same result even after controlling for sex, age, residence time, and work exposure time. Rats daily treated with Mn presented the first evidence that chronic Mn intoxication leads to activity and rest rhythm impairments in another study about metals and the endocrine disruption process [39].

Chuang et al. [55] reported higher UMn associated with decreased cortisol and serotonin levels in workers, important hormones related to the sleep–wake cycle. Serotonin is important in the melatonin synthesis cascade, which exhibits its acrophase in the middle of the night and remains lower during the day. Cortisol has its acrophase in the early hours of dawn and remains lower for the rest of the day. Therefore, metals such as Mn, playing a critical role in neuroendocrine functions, can initiate an adrenergic response or a stimulated cortisol release through the hypothalamic–pituitary–adrenal axis. Furthermore, xenobiotics linked to endocrine disruption may modify the active site of metals in the central nervous system. In addition, endocrine disruption may also be responsible for changes in the sleep–wake cycle rhythm, which involves a reduction in the synthesis of circulating melatonin and loss of serotonin. Thus, the chronic effects of metals and endocrine disruption can lead to two outcomes resulting from the same toxicological process [69]. In the case of Mn, high exposure levels may be associated with morning patterns due to its affinity to cortisol, presenting its acrophase in the early morning, which, in turn, may lead to an advancement of the waking phase in exposed individuals.

### 4.5. Urinary Benzene and Toluene Level Assessments

#### 4.5.1. According to *PER3* Gene VNTR Polymorphisms

Higher levels of non-metabolized benzene and toluene were observed in individuals presenting the afternoon chronotype. For many authors, those volatile organic compounds are toxic even at low concentrations [18,19,20,21,22]. Exogenous substances or their mixtures alter endocrine system function and behave as endocrine disruptors [30,70]. On the other hand, the central nervous system responds quickly to toluene and benzene toxicity [23,71,72]. Thus, both systems regulate and control all human body functions, including melatonin synthesis and hormone level alterations. Consequently, toluene and benzene exposure can lead to sleep disturbances [10,13,24,29].

#### 4.5.2. According to *PER3* Gene VNTR Polymorphisms and Influence Zone

Residents in influence zone 2 (far from the slag) exhibited the highest levels of unmetabolized benzene and toluene in their urine compared to those in influence zone 1, about 2-fold higher for benzene. The location of the collection points can explain this fact. Influence zone 2 was close to the main avenue of the condominium, which delimits it and makes its use mandatory. As it is the main street, the traffic of motor vehicles is high and, consequently, the burning of fossil fuels, mainly diesel and gasoline, is also elevated. Other studies confirmed the same results and proved the great influence of vehicular traffic on the levels of those contaminants in the environment. However, a concrete block factory located next to the sampling site, with several trucks carrying those blocks and burning fossil fuels, also contributed to the increase in detected levels [73,74,75,76].

The explanation for the detected values may be in the origin of those contaminants in the atmosphere. Usually, the presence of benzene is attributed to vehicular emissions, while other processes can also emit toluene due to volatilization [77,78]. The emission of those substances can also be attributed to stationary sources such as factories, industries, and gas stations [79,80,81,82]. In the steel industry, by-products such as benzene and toluene are generated in the coking step [83], an important source of widespread environmental exposure, given the volatilizing capacity of these substances.

#### 4.5.3. According to *PER3* Gene VNTR Polymorphisms and Occupational Chemical Exposure

Afternoon chronotype unexposed workers presented 2-fold higher benzene concentrations in urine (µg L^−1^) compared to workers who reported occupational chemical exposure, in contrast with the results reported by Gore and collaborators [4]. In a review, the authors warned that occupational groups might also face exposure risks in addition to resident communities close to industrial activities [13].

The fact that the highest concentrations of urinary benzene were found in unexposed workers can be explained by workers omitting actual exposure for fear of losing their jobs at the steel company or related to the fact that 64% of afternoon chronotype workers reside in the zone of influence 2. It is important to highlight that exposure to benzene and toluene desynchronizes the sleep–wake cycle, affecting the monoaminergic response in brain areas related to sleep, which includes pineal gland function. The monoaminergic system is formed by noradrenaline, serotonin, and dopamine, which act in the modulation and integration of several cortical and subcortical activities, in turn regulating psychomotor activity, mood, appetite, and sleep [29]. Disorders related to chemical substance exposure, including benzene and toluene, are due to serotonergic transmission deficiency. As previously described, the neurotoxic consequences caused by benzene and toluene are no different from toxic metals such as Mn and Pb [10,19,24,39,61,72].

#### 4.5.4. Study Limitations

Our study exhibits limitations that should be considered, such as a cross-sectional design, which allows only associations and long-term data collection (2017–2019), and not causality verifications. Even when applying a randomized sampling, low adherence to the study was noted, as many residents work or are related to workers in the local steel industry and were afraid of losing their jobs. Because of this, non-parametric tests were employed, which made it difficult to compare the results to population data. The data could also not be controlled for working conditions or smoking and drinking habits. Therefore, the findings reported herein may be applicable only to the specific population evaluated in this study.

## 5. Conclusions

To the best of our knowledge, this study is the first involving the investigation of an adult population living in a steel-contaminated area comprising a genetic analysis concerning chronotypes.

Chronobiological, toxic, and carcinogenic consequences were noted in the present study. So, we cannot rule out the hypothesis that Mn, Pb, benzene, and toluene may be responsible for changes in the sleep–wake cycle rhythm, which may have influenced the different chronotypes found in the steel residue-exposed population. Therefore, the condominium built on a steel residue-contaminated area is of concern, requiring the continuous monitoring of the local population. In addition, the *PER3* gene polymorphisms or the phenotypic chronotype can be used as a marker for endocrine changes and help in public policies related to the local health system and monitored environmental areas.

## Figures and Tables

**Figure 1 ijerph-20-04760-f001:**
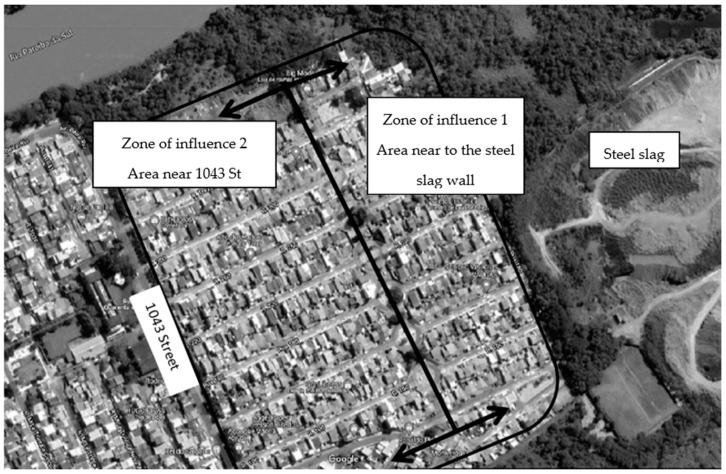
Delimitation of the influence zone concerning benzene and toluene exposure in the Volta Grande IV condominium.

**Figure 2 ijerph-20-04760-f002:**
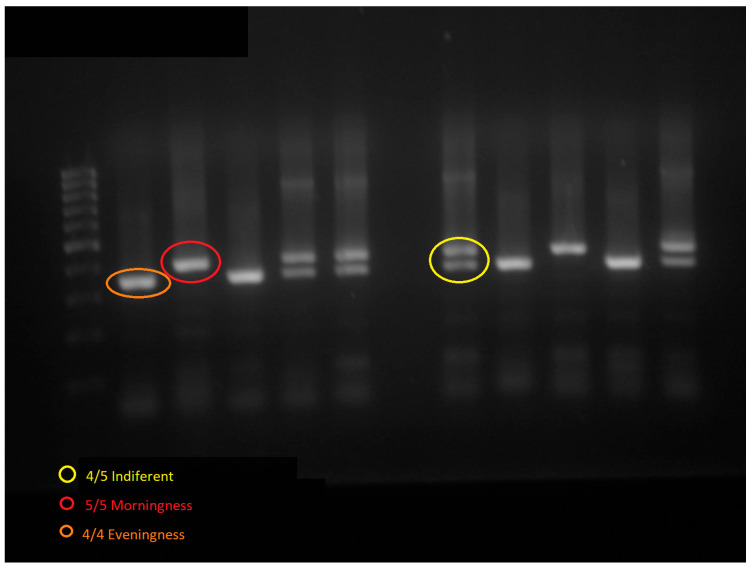
Image from the electrophoresis gel. PCR products corresponding to *PER3*^4/4^ (evening), *PER3*^5/5^ (morning), and *PER3*^4/5^ (intermediate) chronotypes.

**Figure 3 ijerph-20-04760-f003:**
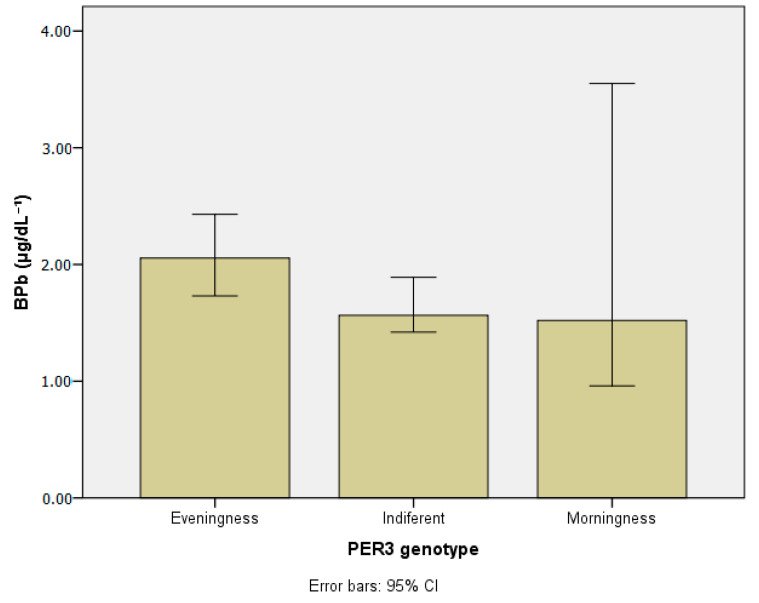
Lead in blood (µg/dL^−1^) according to *PER3* gene VNTR polymorphisms.

**Figure 4 ijerph-20-04760-f004:**
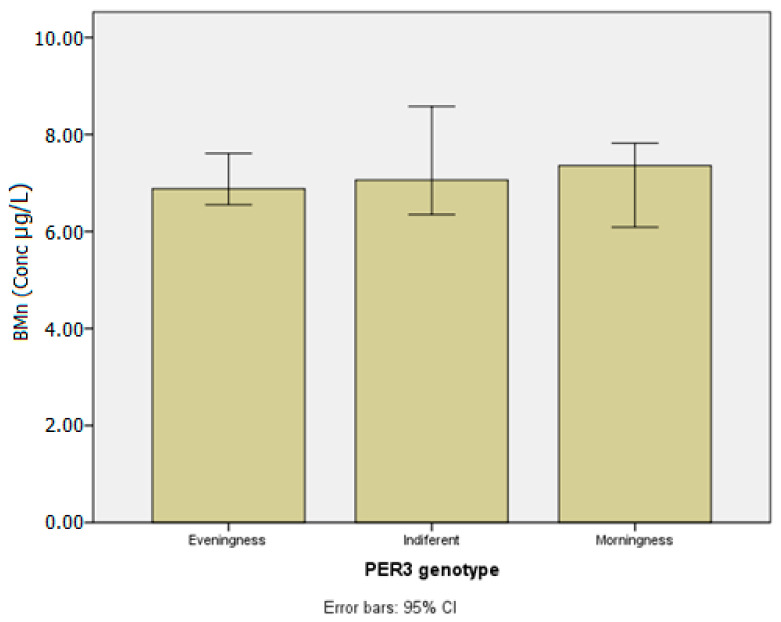
Manganese in blood (µg/L) according to *PER3* gene VNTR polymorphisms.

**Figure 5 ijerph-20-04760-f005:**
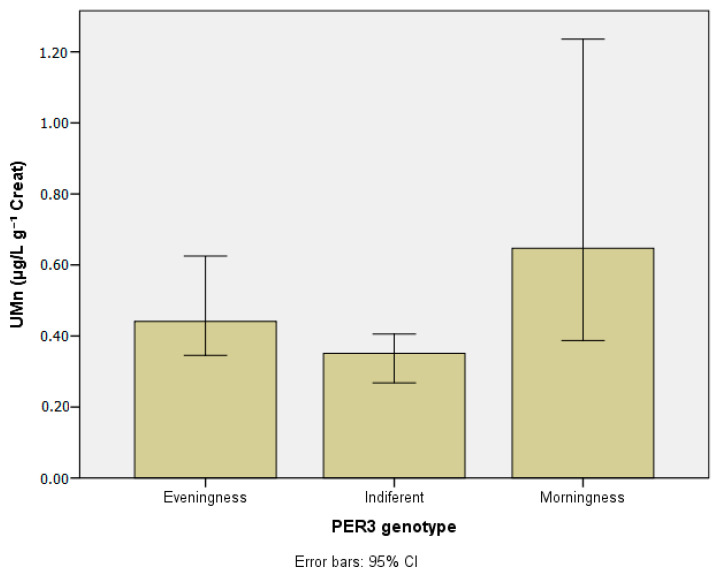
Manganese in urine (µg g^−1^ creat) according to *PER3* gene VNTR polymorphisms.

**Figure 6 ijerph-20-04760-f006:**
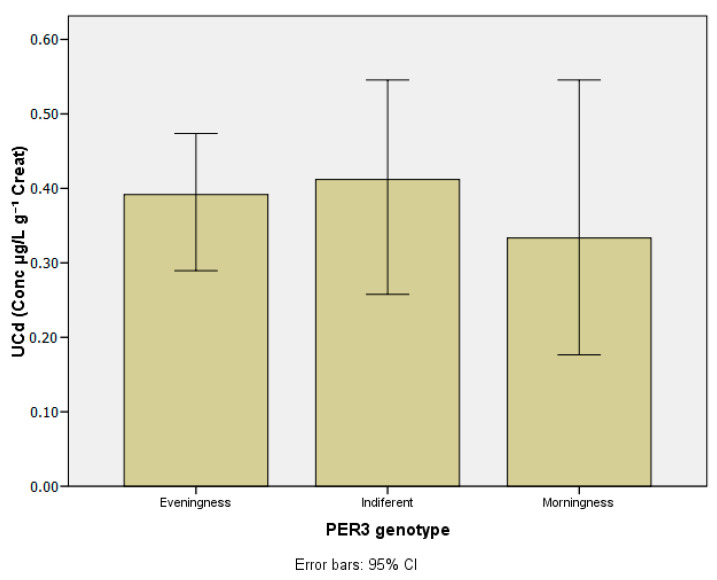
Cadmium in urine (µg g^−1^ creat) according to *PER3* gene VNTR polymorphisms.

**Figure 7 ijerph-20-04760-f007:**
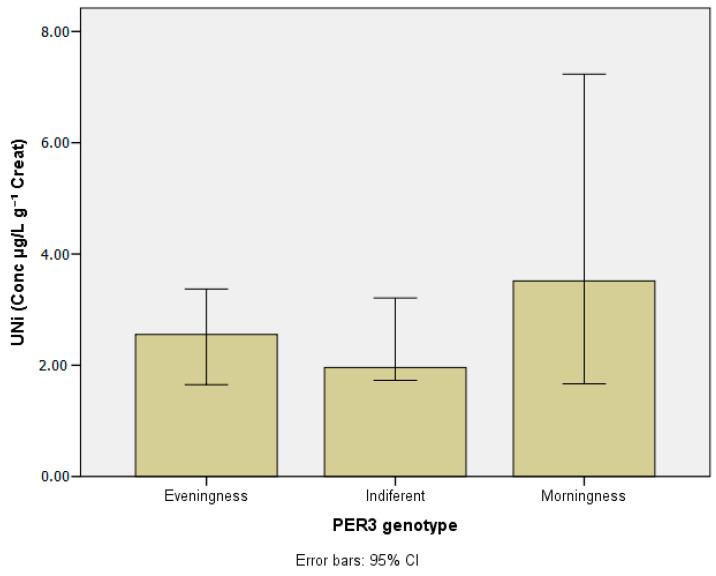
Nickel in urine (µg g^−1^ creat) according to *PER3* gene VNTR polymorphisms.

**Figure 8 ijerph-20-04760-f008:**
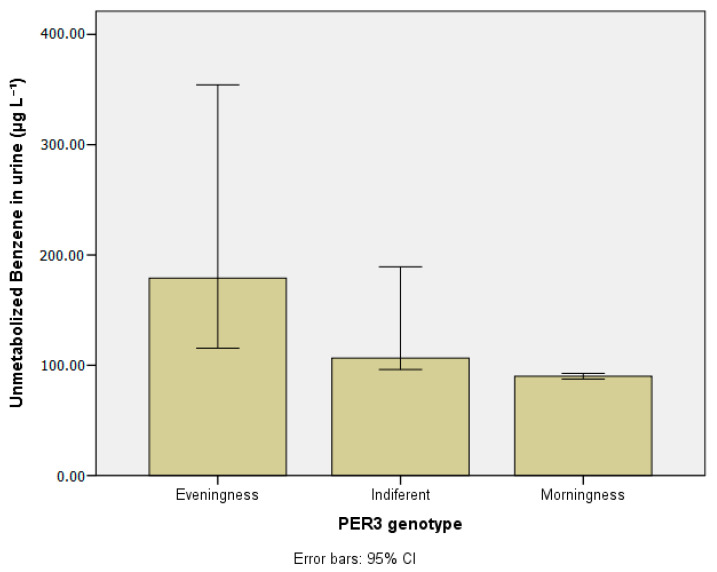
Unmetabolized benzene in urine (µg L^−1^) according to *PER3* gene VNTR polymorphisms.

**Figure 9 ijerph-20-04760-f009:**
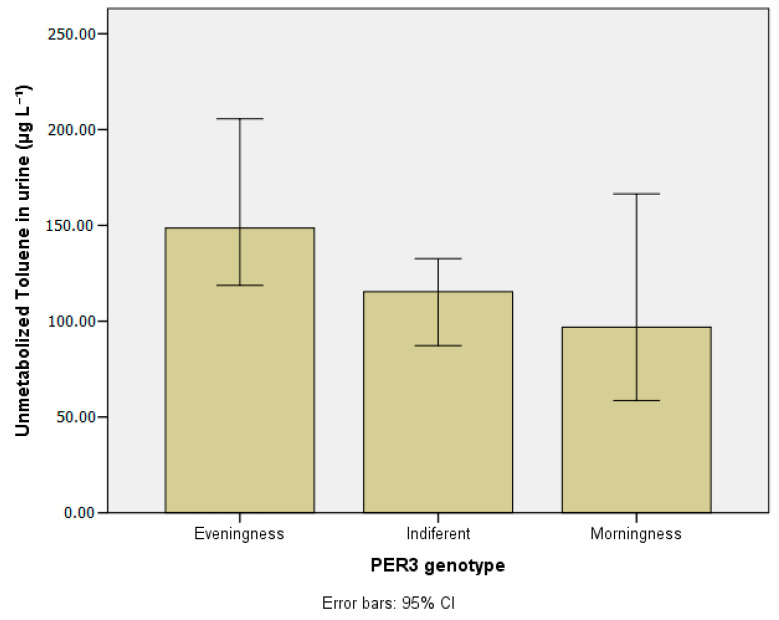
Unmetabolized toluene in urine (µg L^−1^) according to *PER3* gene VNTR polymorphisms.

**Table 1 ijerph-20-04760-t001:** Sleep patterns according to *PER* 3 VNTR polymorphism.

Sleep Patterns/ *PER3*	Evening			Indifferent			Morning		
	P_25_	Median	P_75_	P_25_	Median	P_75_	P_25_	Median	P_75_
Sleep latency (min)	5 min	15 min	30 min	8 min	15 min	40 min	7 min	10 min	20 min
Bedtime (PM)	10:00	11:00	12:00	10:00	11:00	12:00	11:00	11:00	11:45
Wake-up period (AM)	6:00	6:30	7:00	6:00	6:30	7:30	6:00	6:00	7:50
Sleep duration (h)	7 h	8 h	8:30 h	7 h	8 h	9 h	6:15 h	7 h	8 h

**Table 2 ijerph-20-04760-t002:** Sleep complaints according to PER 3 VNTR polymorphism.

Variable	Categories	Chronotype		
		Evening	Indifferent	Morning
		N (%)	N (%)	N (%)
Insomnia	No	66 (54)	43 (35)	14 (11)
	Yes	10 (32)	18 (58)	3 (10)
Excessive sleepiness	No	69 (52)	49 (37)	15 (11)
	Yes	9 (36)	15 (60)	1 (4)
Quality of sleep	No	56 (47)	49 (41)	14 (12)
	Yes	23 (55)	16 (38)	3 (7)
Mood	No	60 (50)	48 (40)	11 (9)
	Yes	17(43)	17 (43)	4(15)

**Table 3 ijerph-20-04760-t003:** Lead and manganese in blood, and cadmium, nickel, and manganese in urine according to *PER3* gene VNTR polymorphisms.

Metal/Chronotype	Evening			Indifferent			Morning			Kruskal–Wallis χ^2^	*p*-Value
	P_25_	Median	P_75_	P_25_	Median	P_75_	P_25_	Median	P_75_		
BPb (µg dL^−1^)	1.27	2.05	2.72	1.27	1.56	2.13	1.00	1.52	2.00	5.62	0.06
BMn (µg L^−1^)	5.85	6.88	8.84	5.58	7.06	9.61	5.66	7.36	8.39	0.49	0.79
UCd (µg g^−1^ Creat)	0.23	0.39	0.81	0.19	0.41	0.92	0.16	0.33	0.66	0.61	0.74
UNi (µg g^−1^ Creat)	1.24	2.55	4.40	1.38	1.96	4.64	1.43	3.51	7.62	1.41	0.49
UMn (µg g^−1^ Creat)	0.28	0.44	0.92	0.22	0.35	0.49	0.37	0.64	1.79	9.16	0.01 *

* Statistically significant.

**Table 4 ijerph-20-04760-t004:** Lead in blood (µg dL^−1^) according to *PER3* VNTR polymorphism and stratified by age group, drinking habits, and chemical exposure at work.

Variable	Categories	Evening			Indifferent			Morning		
		P_25_	Median	P_75_	P_25_	Median	P_75_	P_25_	Median	P_75_
Sex	Female	1.36	1.85	2.19	1.31	1.47	1.73	096	1.18	1.27
	Male	1.83	2.27	2.70	1.52	1.66	2.13	1.61	1.90	3.03
Age group	<21	1.15	1.14	0.09	1.39	1.54	0.26	-	-	-
	22–50	1.92	1.96	0.65	1.59	1.59	0.57	1.78	1.52	1.26
	>50	2.55	2.26	1.66	1.85	1.60	0.82	1.79	1.54	0.95
Smoking habit	No	1.29	2.01	2.29	1.29	1.47	1.68	1.03	1.49	1.83
	Former smoker	1.86	2.40	2.92	1.39	2.45	2.70	-	-	-
	Yes	1.91	2.43	2.72	-	-	-	1.35	1.52	3.10
Current drinking habits	No	1.93	2.04	0.72	1.51	1.42	0.66	1.15	1.07	0.48
	Yes	2.96	2.60	2.06	1.84	1.58	0.81	2.45	2.00	1.20
Chemical exposure at work	No	2.17	2.15	1.22	1.65	1.54	0.72	1.76	1.80	0.92
	Yes	2.49	1.91	1.91	1.79	1.82	1.64	0.66	1.26	0.37

**Table 5 ijerph-20-04760-t005:** Manganese in blood (µg L^−1^) according to *PER3* gene VNTR polymorphisms stratified by sex, age group, smoking and drinking habits, and chemical exposure at work.

Variable	Categories	Evening			Indifferent			Morning			Test	*p*-Value
		P_25_	Median	P_75_	P_25_	Median	P_75_	P_25_	Median	P_75_		
Sex	Female	5.91	6.89	8.87	6.01	8.90	11.10	6.70	7.71	9.58	(W) = −2.39	0.02 *
	Male	5.74	6.78	8.77	5.23	6.41	8.52	4.90	5.59	7.30		
Age group	<21	8.64	9.61	10.43	6.41	7.85	9.26	-	-	-	χ^2^ = 6.81	0.03 *
	22–50	6.20	7.58	8.56	6.46	8.37	10.48	7.36	7.72	8.96		
	>50	6.21	6.64	7.19	5.73	6.94	8.21	5.64	6.70	7.52		
Smoking habit	No	5.63	6.88	8.72	5.80	7.06	9.19	5.23	7.36	8.96	n.s.	
	Former smoker	5.78	6.48	7.98	4.77	7.52	9.74	7.14	7.42	-		
	Yes	5.10	8.71	13.01	4.61	5.70	7.38	4.90	7.78	-		
Current drinking habit	No	5.94	6.88	8.88	5.43	6.87	9.63	6.52	7.50	9.14	n.s.	
	Yes	5.02	6.55	8.15	5.90	7.52	8.96	4.91	7.14	7.78		
Chemical exposure at work	No	5.86	6.74	8.23	5.57	7.52	10.13	5.85	7.36	9.27	n.s.	
	Yes	5.76	7.64	8.94	5.61	6.94	8.58	4.90	5.49	-		

* Statistically significant; n.s. not significant.

**Table 6 ijerph-20-04760-t006:** Linear regression of the independent factors associated with chronotypes ^1^.

Coefficients	Non-Standardized Coefficients		Standardized Coefficients	t	Sig.	95.0% Confidence Interval for B	
Model	B	Standard Error	Beta			Lower Limit	Upper Limit
(Constant)	44.599	1.604		27.813	0.000 *	41.403	47.795
Mn (µg g^−1^ Creat)	0.798	0.224	0.387	3.564	0.001 *	0.352	1.244
Sex	1.137	0.571	0.215	1.990	0.050 *	−0.001	2.276
Age	−0.033	0.027	−0.137	−1.198	0.235	−0.087	0.022
Residence time	−0.048	0.053	−0.101	−0.892	0.375	−0.154	0.059
Exposure time at work	0.018	0.029	0.071	0.602	0.549	−0.041	0.076
**Model Summary ²**							
**Model**	**R**	**R Square**	**Adjusted R Square**		**Standard Error of the Estimate**		
	0.449 ²	0.202	0.147		2.450		

^1^ Dependent Variable: *PER3* Genotype. ^2^ Predictors: (Constant), time and exposure at work, Mn (µg g^−1^ Creat) sex, residence time, age. * statistically significant.

**Table 7 ijerph-20-04760-t007:** Concentrations of unmetabolized benzene and toluene in urine (µg L^−1^) according to *PER3* gene VNTR polymorphism.

Benzene and Toluene/Chronotype	Evening			Indifferent			Morning			Friedman Test	*p*-Value
	P_25_	Median	P_75_	P_25_	Median	P_75_	P_25_	Median	P_75_		
Unmetabolized benzene (µg L^−1^)	128.88	179.04	336.66	99.67	106.57	169.87	87.42	90.02	-	21.00	<0.01 *
Unmetabolized toluene (µg L^−1^)	80.68	148.58	274.44	79.03	115.36	197.42	66.26	96.89	161.45	91.66	<0.01 *

* Statistically significant.

**Table 8 ijerph-20-04760-t008:** Unmetabolized benzene and toluene in urine (µg L^−1^) according to *PER3* gene VNTR polymorphism and zone of influence.

Variables	Category	Evening			Indifferent			Morning			Kruskal–Wallis Test	*p*
		P_25_	Median	P_75_	P_25_	Median	P_75_	P_25_	Median	P_75_		
Benzene (µg L^−1^)	Influence zone 2	233.03	387.28	510.54	104.00	128.53	179.57	-	-	-	9.66	0.05 *
	Influence zone1	115.51	152.24	214.38	-	-	-	-	-	-		
Toluene (µg L^−1^)	Influence zone 2	114.22	164.00	293.69	68.55	105.63	175.15	56.22	120.91	166.47	5.71	0.34
	Influence zone 1	66.68	126.58	344.45	78.50	119.59	195.92	78.36	84.54	418.15		

* Statistically significant.

**Table 9 ijerph-20-04760-t009:** Unmetabolized benzene and toluene in urine (µg L^−1^) according to *PER3* gene polymorphisms and occupational exposure to chemicals.

Variables	Category	Evening			Indifferent			Morning			Kruskal–Wallis Test	*p*
		P_25_	Median	P_75_	P_25_	Median	P_75_	P_25_	Median	P_75_		
Chemical Exposure at Work Benzene (µg L^−1^)	No	179.04	219.91	387.28	103.14	106.57	-	-	-	-	11.20	0.02 *
	Yes	65.78	115.51	155.03	-	-	-	-	-	-		
Chemical Exposure at Work Toluene (µg L^−1^)	No	80.82	142.60	271.45	86.61	127.23	207.09	58.60	84.54	166.47	6.98	0.14
	Yes	66.68	163.51	281.31	56.22	85.15	116.86	-	-	-		

* Statistically significant.

## Data Availability

The datasets generated and analyzed during the current study are not publicly available since personal data from subjects are protected according to the Ethics Committee but may be available from the corresponding author in aggregate form on reasonable request.

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
