# Peer review of "Influence of Environmental Exposure to Steel Waste on Endocrine Dysregulation and PER3 Gene Polymorphisms"

_ijerph, 2023, doi:10.3390/ijerph20064760_

Round 1

Reviewer 1 Report

This manuscript evaluated the association between environmental exposure to the following 15 chemical substances cadmium (Cd), lead (Pb), nickel (Ni), manganese (Mn), benzene (BZN), and 16 toluene (TLN), and PER3 gene polymorphisms, according to chronotype in a population residing in 17 a steel residue-contaminated area. The participants answered questions on socio-demographic, life style, diseases history and current health conditions. They were also given the Pittsburgh Sleep Quality Index (SQI) scale. The research is very important. However, the manuscript contains multiple grammatical errors. The background is unclear and the results in some cases unclearly presented.

Detailed Comments

Introduction

The motivation for this study is not clearly articulated. The literature review in the background is not relevant to the subjects. The authors should discuss evidence on the subject and clearly established the knowledge gap in the area.

L35-39, 46-50, 53-55, 195: long and unclear sentences

L60: Its refers to .................

L96: That was?

L99-102: How the sampling was done needs to be clearly explained. What informed your sample size? Was the number picked arbitrarily. Randomization and random selection are not the same. Please correct.

L122-3: Explain why Cr, Ni, and Cd were determined in urine, Mn in urine and blood, and Pb in blood

L140-141: "controlled by potential confounders. "

L141-142: You cannot control for both smoking and drinking habits but you can control for one.

L87/88: I am not too sure what the authors meant here. Acute respiratory infections (ARIs) are classified as upper respiratory tract infections (URIs) or lower respiratory tract infections (LRIs). What do they mean by saying 'About 459,000 of these deaths and 25.9 million of disability adjusted life years (DALYs) were due to acute

respiratory infections (ARIs) and Acute Lower Respiratory Infections (ALRI)'

In Table 4 and 5, the authors stratified BPb and BMn by participant characteristics. However, the variables that they stratified on are not the same for each table.

Why didn't the authors stratified for UCd, UNi, and UMn in the Tables.

What do the authors mean by error bars in Fig. 1

Table 6: The authors should re-do the analysis considering smoking + other confounders separately and alcohol+other confounders separately. And decide on which one to present. This should be done in consultation with a biostatistician

Why was the regression analysis done for only Mn. What about the other metals?

L244-245: very confusing. pls clarify this. The p-value is not for the evening.

L250-255: not too sure if the information conveyed here, is a result. description of the condominium fits well in the methods section.

Section 3.5.3: pls modify this sub-title

L270-273: This is not what the p-value means

L277,278,280 and elsewhere in the text: the use of “association” is confusing

L297-299: what does this mean? how do you interpret this?

L316: Is there any mechanism for this observation.

L341-342: Not too sure of the relevance of this sentence here

L347: Statistically significance mean different from what you are saying here. Please revise

L357-358: Please, check how you interpret your p-values

The strengths and limitations of the study should be articulated. What is this study contributing and what are/is the Global health significance of this study? these should be clearly articulated

Author Response

The authors appreciate the recommendations given by the reviewer, which improved the article.

The file attached contains the answers, item by item.

Reviewer 2 Report

Coutinho et al. evaluated the association between environmental exposure to the cadmium, lead, nickel, manganese, benzene, and toluene, and PER3 gene polymorphisms, according to chronotype in a population residing in contaminated area. I suggest the article for publication after minor revision.

1.    Have you done study in a population residing in non-steel residue-contaminated area? If yes, could you discuss comparative data?

2.     How can you confirm that live next to a steel waste dump influence on sleep-wake cycle synchronism rather than some other personal factors?

3.     Please move Figure 2 to the Supporting information.

4.     Please provide graphs for the rest studies elements in the Figure 1 (lead and manganese in blood; cadmium and nickel in urine according to PER3 gene polymorphisms).

5.     What is error bar on the Figure 1? Is that standard deviation?

6.     Please provide graphs that show concentrations of unmetabolized benzene and toluene in urine according to PER3 gene polymorphism.

Author Response

(The authors gave the same response as above.)

Reviewer 3 Report

Your proposal denotes a great work and I consider that it has been difficult to gather and expose a lot of data, however I recommend that you improve the role of the PER3 gene, for this I recommend the following:

1- follow the HGNC nomenclature guidelines during the writing of the gene name.

2. Define all acronyms before mentioning them (including in the abstract).

3. Introduction.-  Please define the different Chronotypes and characteristics related with Circadian clock and here influence in endocrine function. The researcher should justify why  decides to relate PER3 polymorphisms and explain the possible metabolic pathways that will be modified depending on their variants and each contaminant.

4. Materials and Methods.- 4. The authors refer to use the methodology published by Pereyra, however this author only mentions that he used the methodology published by Ebisawa in 2001 for the molecular analysis of polymorphisms in PER3. However, Ebisawa only publishes the primer's, but not the conditions of amplification nor those of digestion of the fragments to classify the wild, heterozygous and mutated genotypes, and he also uses sequencing to confirm the amplification of the polymorphisms, so I suggest the authors to abound in this technique that is fundamental in their results and discussion. It is also necessary the calculations of Hardy–Weinberg equilibrium, pairwise linkage disequilibrium and haplotype frequencies  should be to ensure the stability of the gene pool in the population studied.

And therefore it is necessary to present the allelic and genotypic frequencies of the population and to present a strategy that demonstrates that the fragments analyzed really present the polymorphisms that are intended to be identified.

5. RESULTS. It is important that the authors report the allelic and genotypic frequencies of the polymorphisms studied in PER3 and compare them with the frequencies reported in other populations in order to make comparisons on the effects of industrial and urban pollutants in their population. 

5. Discusion. Line 277.  Linea 277. Which PER3 polymorphism does it refer to? 

Conclusions. lines 436-441, Secondnd paragraph. The author hypothesizes that effect on ACTH and Cortisol, however these hormones were not measured in his population therefore he cannot conclude that they were altered by PER3 polymorphisms and their sleep-wake cycle rhythm.  

Author Response

(The authors gave the same response as above.)

Round 2

Reviewer 1 Report

Do not have any further concerns

Author Response

The authors then express gratitude for the reviewer's time spent on the submitted manuscript.

Reviewer 3 Report

"About this point "It is also necessary the calculations of Hardy-Weinberg equilibrium, pairwise linkage disequilibrium and haploide frequencies should be to ensure the stability of the gene pool in the population study.

I Consider that, the scope of the journal it is the editor's decision whether to include the article or not in its content, but an analysis that is essential for the interpretation of the results and their conclusion cannot be omitted.

Author Response

The authors once more appreciate the given recommendations. The answers to the questions can be found in the attached file.
